# Manifesting Epoxide and Hydroxyl Groups in XPS Spectra and Valence Band of Graphene Derivatives

**DOI:** 10.3390/nano13010023

**Published:** 2022-12-21

**Authors:** Maxim K. Rabchinskii, Vladimir V. Shnitov, Maria Brzhezinskaya, Marina V. Baidakova, Dina Yu. Stolyarova, Sergey A. Ryzhkov, Svyatoslav D. Saveliev, Alexander V. Shvidchenko, Denis Yu. Nefedov, Anastasiia O. Antonenko, Sergey V. Pavlov, Vitaliy A. Kislenko, Sergey A. Kislenko, Pavel N. Brunkov

**Affiliations:** 1Ioffe Institute, Politekhnicheskaya St. 26, 194021 Saint Petersburg, Russia; 2Helmholtz-Zentrum Berlin für Materialien und Energie, Hahn-Meitner-Platz 1, 14109 Berlin, Germany; 3NRC “Kurchatov Institute”, Akademika Kurchatova pl. 1, 123182 Moscow, Russia; 4St. Petersburg State University, Universitetskaya nab. 7–9, 199034 St. Petersburg, Russia; 5Joint Institute for High Temperatures of RAS, Izhorskaya St. 13/2, 125412 Moscow, Russia; 6Skolkovo Institute of Science and Technology (Skoltech), Bolshoy Boulevard 30, bld. 1, 121205 Moscow, Russia

**Keywords:** 2D materials, graphene, functionalization, band structure engineering, electronic structure, derivatization, DFT calculations

## Abstract

The derivatization of graphene to engineer its band structure is a subject of significant attention nowadays, extending the frames of graphene material applications in the fields of catalysis, sensing, and energy harvesting. Yet, the accurate identification of a certain group and its effect on graphene’s electronic structure is an intricate question. Herein, we propose the advanced fingerprinting of the epoxide and hydroxyl groups on the graphene layers via core-level methods and reveal the modification of their valence band (VB) upon the introduction of these oxygen functionalities. The distinctive contribution of epoxide and hydroxyl groups to the C 1*s* X-ray photoelectron spectra was indicated experimentally, allowing the quantitative characterization of each group, not just their sum. The appearance of a set of localized states in graphene’s VB related to the molecular orbitals of the introduced functionalities was signified both experimentally and theoretically. Applying the density functional theory calculations, the impact of the localized states corresponding to the molecular orbitals of the hydroxyl and epoxide groups was decomposed. Altogether, these findings unveiled the particular contribution of the epoxide and hydroxyl groups to the core-level spectra and band structure of graphene derivatives, advancing graphene functionalization as a tool to engineer its physical properties.

## 1. Introduction

Engineering two-dimensional materials’ electronic structure has been one of the key issues since their isolation, starting with graphene and followed by metal dichalcogenides layers and elemental two-dimensional materials [1,2,3], leveraging the benefits of the materials’ two-dimensional nature. Extensive experimental and theoretical studies are being carried out to approach the management of the materials’ band gap, govern the position of the Fermi level, and arrange the density of states (DOS) in the valence and conduction bands [4,5,6,7,8,9,10,11]. Several strategies have been advanced for these purposes—structural nanopatterning [5,6], the formation of Moiré heterostructures [7,8], doping with pnictogen or chalcogen atoms [9], and chemical functionalization [1,10,11]. The latter approach has appeared to be the most facile one up to date, particularly for graphene with its inherent ease of being modified by chemical groups with the rise of a large family of its chemical derivatives [11,12,13]. Notably, one of the most renowned graphene chemical counterparts, graphene oxide (GO), was known even before the discovery of graphene, although it was named and reviewed as graphitic acid [14].

The chemical modification of graphene benefits by engineering its electronic structure due to performing it at multiple levels, namely: (i) the regulation of its local and absolute π-conjugation degree, which determines the material’s band gap; (ii) the management of the Fermi level and, thus, the *p*- or *n*-doping of the graphene layer owing to the electron-donating or electron-withdrawing effect of the modifying functional groups; and (iii) the introduction of local electron states in the valence or conduction bands, related to energy levels corresponding to the molecular orbitals (*MO*s) of the added moieties. To date, the main results have been achieved within the frame of the first approach. The graphene band gap has been demonstrated to be tuned within the range of 0.05–4.5 eV by means of altering the functionalization of its basal plane with oxygen groups or fluorine [15,16,17]—that is, the synthesis and subsequent chemistry management of GO or fluorographene via thermal or chemical treatment [18,19]. Furthermore, a vast number of publications devoted to doping graphene by means of functionalization with nitrogen-containing groups, such as amines, pyridines, pyrroles, or aromatic moieties, have been published as well [20,21].

Conversely, almost no attention has been paid to the question of the effect of molecular-related states of the introduced functional groups on the DOS of graphene. Only a generalized interpretation of the oxygen-related states has been given [16,22,23]. One of the reasons is commonly the non-stoichiometric and dynamic composition of the introduced groups, the flip side of the current state of chemical modification of graphene and two-dimensional materials [1,12,14]. These factors, as well as the peculiarities of the electrostatic interaction between the groups, hinder the identification of the molecular-related states, corresponding to a particular introduced functional group. At the same time, these findings are crucial for both advancing the design of the Moiré heterostructures and the facile application of graphene derivatives in the field of heterogeneous catalysis, gas sensing, and biosensing [24,25,26], for which the proximity of the energy levels of the substrate and target chemical is of prime importance [27,28,29,30]. The accurate management of the graphene’s chemistry and electronic structure is also of prime interest for its optoelectronic applications as well as for the designing and fabrication of functional nanocomposites [31,32,33].

Recently, we have revealed the appearance of a set of localized molecular-related states in graphene’s VB upon its derivatization by carboxyl groups and ketones [34], as the most stable oxygen groups in GO. In turn, no substantial effect of basal-plane thiol groups on the DOS of the graphene layer has been identified for the thiolated graphene [35]. However, no insight into the localized states introduced by hydroxyls and epoxides, abundant in GO but sufficiently unstable to external factors, has been proposed yet. Several works have demonstrated the synthesis of epoxidated (E-xy) and hydroxylated (H-xy) graphenes covered predominantly by one of these groups, but without delving into the identification and designation of the electronic states related to the presence of hydroxyls and epoxides [19,36,37]. The lack of attention to this question is due to the challenging quantitative examination of the H-xy and E-xy graphenes by means of core-level spectroscopy methods, namely X-ray photoelectron spectroscopy (XPS) and X-ray absorption spectroscopy (XAS). A minor difference in the binding energies of the corresponding components in the C 1*s* and O 1*s* X-ray photoelectron spectra hinders the estimation of the absolute and relative concentration of hydroxyls and epoxides. Combined with the absence of an unequivocal equation of the resonances in the X-ray absorption spectra to hydroxyl and epoxide groups, all this challenges further interpretation of the effect of these functionalities on the band structure of the material. Only solid-state ^13^C nuclear magnetic resonance (NMR) spectroscopy grants the unequivocal discernment of hydroxyls and epoxides and the derivation of their relative concentration [38], although the applicability of this method is complicated and limited by issues in measuring conductive materials.

Herein, taking advantage of our recent advances in XPS and theoretical studies of other graphene derivatives [34,35], the fingerprints of epoxide and hydroxyl groups in the C 1*s* XPS and C K XAS spectra, as well as the molecular-related states in the valence band (VB), are revealed via the systematic study of diversely oxidized graphene layers by a set of spectroscopic methods complemented by density functional theory (DFT) modeling. Based on the comparative study of the XPS spectra with the NMR and infrared (IR) spectroscopy data, the distinctive spectral components related to the epoxide and hydroxyl groups were discerned and characterized in the C 1*s* spectra. Furthermore, the corresponding fingerprints in the C K XAS spectra were identified. Given this data on the chemistry of the studied graphene derivatives, contributions of the hydroxyl and epoxide groups to the VB structure were decomposed both experimentally and theoretically. The origin of each molecular state in the epoxide and hydroxyl groups was revealed by the theoretical calculations of both the DOS of the whole CMG layer and its projection solely on the modifying group. As a net result, these findings unveiled the particular contribution of epoxide and hydroxyl groups to the core-level spectra and band structure of graphene derivatives, advancing chemical derivatization as a tool to engineer their physical properties.

## 2. Materials and Methods

### 2.1. Materials

An aqueous dispersion of GO was purchased from Graphene Technologies (Moscow, Russia, www.graphtechrus.com (accessed on 30 October 2022)). Sodium hydroxide (NaOH) and hydrochloric acid (HCl) were purchased from Merck KGaA (Darmstadt, Germany).

All the organic solvents used in this work were purchased from Vecton Ltd. (Saint-Petersburg, Russia). All the chemicals were of analytical purity grade and were commercially available. The materials were used as received without any additional purification.

### 2.2. Synthesis of E-xy Graphene, H-xy Graphene, and rGO

H-xy and E-xy graphene were derived by the liquid-phase modification of the pristine GO. In brief, to synthesize H-xy graphene, 50 mL of a GO aqueous suspension of 0.1 wt.% concentration was poured into a Teflon reaction vessel and mixed with NaOH while stirring to reach a *pH* value of 8.5. The pH values of the suspensions were evaluated with a Fisher Scientific Accumet Basic AB15 pH meter (Thermo Fisher Scientific, Waltham, MA, USA). Thus, the acquired reaction mixture was stored under heating at *T* = 50 °C for 48 h in the air, while continuously stirring. Upon the end of heating, the mixture was cooled down to room temperature (*RT*) and copiously washed by means of centrifugation (Sigma S-16 centrifuge, Germany) at 18,200× *g* for 15 min with sequential rinsing of the obtained sediment by distilled water (30 mL). The ascribed washing procedure was repeated five times to obtain an aqueous suspension of H-xy graphene platelets.

E-xy graphene was derived from GO by substituting aqueous media with dimethyl sulfoxide (DMSO) via sequential centrifugation and the following treatment with an HCl solution. In brief, 50 mL of a GO aqueous suspension of 0.1 wt.% concentration was centrifuged at 18,200× *g* for 15 min with sequential rinsing of the obtained sediment with DMSO, repeating the ascribed procedure 5 times to almost completely eliminate water from the GO. Afterward, 10 mL of 0.05 M HCl was poured into the GO suspension in DMSO, and the Teflon reaction vessel was placed in the oven to be heated to 50 °C for 48 h. At the expiry time, the mixture was cooled down to *RT* and copiously purified analogously to the H-xy graphene, except for replacing the water with ethanol.

The rGO layers were fabricated by the thermal annealing of GO at *T* = 600 °C in an ultra-high vacuum chamber (*p* = 10^−9^ Torr) for 3 h: first, 1.5 h to reach the temperature, and secondly, 1.5 h at the maintained temperature.

### 2.3. CMG Characterization

The composition of functional groups in the synthesized graphene derivatives was examined via a set of core-level techniques and spectroscopic methods, namely XPS, XAS, Fourier-transform infrared (FTIR), and NMR spectroscopies.

The survey, C 1*s*, and VB XPS spectra along with C *K*-edge XAS spectra were collected at the ultra-high vacuum experimental station of the Russian–German beamline of the electron storage ring BESSY-II at Helmholtz-Zentrum Berlin (HZB) [39]. Prior to the measurements, the samples were evacuated down to a pressure of *p* ~ 10^−9^ Torr for 20 h at room temperature (ca. *T* = 25 ± 2 °C) to remove the adsorbates. The measurements were performed in three distinct spots of the sample, of ca. 200 × 100 µm in size, to consider the possible spatial heterogeneity of the materials’ chemistry. The difference in the C 1*s* and VB spectra did not exceed 5%, thus verifying the pertinence of the acquired data. For further processing, the recorded spectra were averaged.

The survey spectra were collected with the excitation energy of *hv* = 736 eV. Conversely, the C 1*s* and VB spectra were acquired at the excitation energies of *hv* = 400 eV and *hv* = 130 eV, respectively. For the latter case, the cross-sections of the 2*p* and 2*s* states were almost equal, allowing the precise derivation of the effect of these states on the structure of the material’s VB. At the same time, the choice of an excitation energy of *hv* = 400 eV for the C 1*s* spectra guaranteed that the probing depths for these spectra and VB ones almost coincided. As a net result, the C 1*s* data accurately reflected the impact of the functional groups on the measured DOS in the VB.

To extract quantitative information on the chemical composition of the examined materials, the collected C 1*s* spectra were deconvoluted into a sum of the Shirley background and five model functions by means of CasaXPS@ software (version 2.3.16Dev52, Casa Software Ltd.). Four functions related to the oxygen functional groups were simulated by symmetrical Gaussian–Lorentzian lineshape functions (GL(*n*)), where parameter *n* was chosen as equal to 90 in our case. The full width at half maximum (FWHM) for all functions was chosen not to exceed 1.8 to adequately restrict the possible level of deviations in the binding energy of oxygen functionalities due to their interaction. The fifth function, associated with the contribution of carbon atoms occupying nonfunctionalized sites of graphene lattice (peak C=C), was simulated using a special CasaXPS function LF (α, β, w, m) (LF(0.8, 1.35, 50, 300)), representing a Lorentzian asymmetric lineshape with tail damping [40], which sufficiently described the asymmetric shape of the C 1*s* line corresponding to domains of the π-conjugated carbon atoms of the graphene network.

The C *K* near-edge X-ray absorption fine-structure spectra were collected in the Auger electron yield (AEY) mode; that is, the collected spectra reflect how the yield of C*KKV* Auger electrons with a kinetic energy of ~250 eV depends on the energy of the SR photons irradiating the respective sample. The angle between the incident beam and the surface normal of the sample was equal to the “magic” angle, α = 54.7°, providing a nearly equal excitation of the π and σ states. The as-recorded XAS spectra were normalized and smoothed by following a processing algorithm described elsewhere [41].

The FTIR spectra were acquired using an Infralum-08 spectrometer (InfraLUM, Russia) equipped with an attenuation of total reflectance attachment. For the XPS, XAS, and FTIR measurements, films of the E-xy graphene, H-xy graphene, and rGO were placed over Si wafers by the drop-casting of 25 μL of the corresponding aqueous suspension, of 5 × 10^−3^ wt.% concentration, with subsequent drying at room temperature (ca. 25 °C) for 12 h. In the case of rGO, the GO suspension was deposited with the subsequent thermal reduction according to the procedure presented in Section 2.2. Owing to the equality of the deposition protocol for all the samples, the thickness of the deposited films was of the same order and was estimated to be ca. 100–120 nm. This fact assumes the feasibility of the FTIR spectra analysis in terms of comparing the relative intensity of the same absorption bands.

NMR spectra for the ^13^C nuclei were collected in the magic angle spinning cross polarization pulse sequence (CPMAS) mode with a magnetic field of 9.4 T and a spinning rate of 8 kHz. Prior to the measurements, the samples were milled from the films into a powder and subsequently loaded into 4 mm-diameter zirconia rotors. To achieve an optimal signal-to-noise ratio, the number of scans for each experiment was chosen to be 8192. All the measurements were carried out at room temperature.

### 2.4. DFT Modeling

DFT modeling was carried out using supercomputers at the Joint Supercomputer Center of the Russian Academy of Sciences (JSCC RAS) [42] and the Zhores HPC cluster at Skoltech [43]. The electronic structure of H-xy and E-xy graphene was simulated with the use of periodical DFT calculations, as implemented in the VASP package [44]. A local density approximation (LDA) exchange-correlation functional with an ultrasoft pseudopotential was used. A Monkhorst–Pack *k*-mesh [45] of 3 × 3 × 1 was set during the ionic optimization, while for static self-consistent calculations, the Brillouin zone sampling was increased to 15 × 15 × 1 k-points. To carry out ionic optimization, a conjugate gradient algorithm was employed with the ionic and electronic convergence criteria set to equal 10^−3^ eV and 10^−4^ eV, respectively. In self-consistent calculations, the electronic convergence criterion was 10^−5^ eV. At the same time, the plane-wave cutoff energy was set as 500 eV in all calculations.

For the calculation of the electron structure of the E-xy and H-xy graphene derivatives, model systems based on a graphene supercell with the size of 19.7 Å × 19.7 Å in the XY and an angle between the X and Y axes of 60° were employed. Three-dimensional periodical boundary conditions were used with 15 Å of vacuum along the *Z*-axis to minimize the interaction between periodical images. Both models were fully optimized. The acquired E-xy and H-xy graphene supercells are displayed in Figure 1.

As test calculations, the GGA, PAW, PBE, and LDA ultrasoft levels of approximations were used and compared. LDA was shown to be more accurate at predicting bond lengths during geometry optimization, with no significant differences in the calculated DOS for the modeled systems. Thus, only the results with LDA are presented.

## 3. Results

### 3.1. FTIR and NMR Studies

Figure 2a exhibits the FTIR spectra of E-xy and H-xy graphenes, with a set of overlapping bands corresponding to the oxygen functionalities decorating the basal plane and edges of the graphene layer. Both spectra were characterized by the presence of a broad hump in the range of *v* = 3000–3700 cm^−1^,comprising a set of absorption bands of the O–H stretching vibrations in hydroxyl groups, carboxyls, and mainly water molecules [46]. The high number of the latter was also reflected by the appearance of an intensive narrow absorption band at *v* = 41,620 cm^−1^, attributed to the bending vibrations of the water molecules [47]. Notably, both absorption bands at *v* = 1620 cm^−1^ and *v* = 3000–3700 cm^−1^ were of enhanced intensity in the case of H-xy graphene, asserting a higher amount of adsorbed water in this material due to its higher hydrophilicity.

Besides water-related absorption bands, both spectra comprised almost equally pronounced absorption bands at *v* = 1050 cm^−1^, *v* = 1280 cm^−1^, and *v* = 1720 cm^−1^, corresponding to edge-located hydroxyls (denoted hereinafter as C-OH(e)), ketone groups, and carboxyls, respectively [48]. Conversely, the absorption band at *v* = 1105 cm^−1^ and *v* = 1365 cm^−1^ related to basal-plane hydroxyls (denoted hereinafter C-OH(b)) on one hand, and the absorption band at *v* = 1225 cm^−1^ of epoxide groups on the other [49], substantially differed in their intensity, with the first ones being dominant in the H-xy graphene and the latter one being distinct in the spectrum of E-xy graphene. Thus, the FTIR spectra implied the successful synthesis of E-xy and H-xy graphene with the dominance of either the epoxide or hydroxyl group.

This assertion was further verified by the ^13^C NMR solid-state spectra of the E-xy and H-xy graphenes displayed in Figure 2b. The acquired spectra are consistent with the literature data on GO, with the set of peaks at the chemical shifts of *δ*
**=** 58 ppm, *δ*
**=** 68 ppm, *δ*
**=** 95 ppm, *δ*
**=** 133 ppm, *δ*
**=** 167 ppm, and *δ*
**=** 191 ppm corresponding to hydroxyls, epoxides, ethers (O-C-O), *sp^2^*-hybridized carbon atoms of the unfunctionalized graphene network, carboxyl groups, and ketone groups, respectively [34,50]. Apparently, despite being presented in both materials, the epoxides and hydroxyls differed in their concentration in the E-xy graphene and H-xy graphene, with the dominance of epoxide in the former one and hydroxyls in the latter one. A further quantitative analysis of the spectra by fitting spectral bands at *δ*
**=** 40–80 ppm by Lorentzian functions allowed an estimation of the relationship between the number of epoxide and hydroxyl groups in both derivatives. For the E-xy graphene, the relation was found to be 68% to 32% in favor of epoxides, whereas for the H-xy graphene, these values were 73% to 27% in favor of hydroxyls.

### 3.2. Characterization of E-xy and H-xy Graphene via X-ray Photoelectron Spectroscopy

Given the results of the FTIR and NMR analyses, the H-xy and E-xy graphenes were examined by means of X-ray photoelectron and X-ray absorption spectroscopies. Figure 3a exhibits the survey XPS spectra of the H-xy and E-xy graphenes along with rGO as a reference material. In both graphene derivatives, dominating C 1*s* and O 1*s* core-level signals at the binding energies (*BE*s) of 284.7 eV and 532.5 eV, respectively, were distinguished. Besides, only negligible peaks at the *BE*s of 168.2 eV and 400.1 eV were discerned, indicating the presence of trace amounts of ammonium sulfate adsorbates retained after the GO synthesis and its following derivatization into H-xy and E-xy graphenes. The concentration of ammonium sulfate was estimated to be less than 0.2 at.%, and we assert that it had a negligible effect on the VB of the graphene layer, which was verified by corresponding studies in the subsequent section.

Figure 3b displays the C 1*s* spectra of E-xy graphene, H-xy graphene, and rGO after the deconvolution routine. All spectra comprised a dominant asymmetric C=C line centered at a *BE* of 284.6 eV, which was related to the pristine graphene network of *sp*^2^-hybridized π-conjugated carbon atoms [51]. On the other hand, two peaks with *BE*s of 288.2 eV and 288.9 eV corresponded to edge-located ketone and carboxyl groups, as has been signified by a number of synchrotron studies of GO and other graphene derivatives [52,53]. Moving to the epoxide and hydroxyl groups, the common contribution of carbon atoms participating in these functionalities was modeled by only one Gauss–Lorentz contour (C-OH and C-O-C peak) [35,51,52]. This arose from two factors: (i) a rather large FWHM of all the GO C 1*s* spectral components derived during deconvolution, which as a rule noticeably exceeded 1.0 eV, and (ii) a relatively small difference in the predicted *BE*s of the C-OH and C-O-C components in the C 1*s* spectra, commonly estimated to be less than 0.2–0.3 eV [54,55,56].

Nevertheless, in the case of the dominance of one of these groups, an observable shift in the second maximum of the C 1*s* spectrum was indicated, owing to the redistribution of the relative integral intensity of the corresponding components. This could be clearly seen in the acquired C 1*s* spectra of the E-xy and H-xy graphene, with the peak at the *BE*s of 286.1–288.1 eV moving to lower *BE*s, from 287.0 eV to 286.7 eV, upon transition from the dominance of epoxides (E-xy graphene) to the enhanced relative concentration of hydroxyls (H-xy graphene), given the discussed FTIR and NMR data. Considering this fact, *BE*s of 287.0 eV and 286.7 eV for the spectral components of the epoxide and hydroxyl groups, respectively, were accepted.

To justify the proposed deconvolution procedure, data on the relative concentration of all oxygen functionalities were acquired via common quantitative processing of the integral intensities of the corresponding spectral features in the C 1*s* spectra. The respective values of these groups’ relative concentrations are collected in Table 1. According to these data, the relative concentrations of the epoxide and hydroxyl groups in the E-xy and H-xy graphene were 19.5 at.%/3.7 at.% and 3.2 at.%/21.7 at.%, respectively. These values correspond to the ratio of epoxide and hydroxyl groups in the E-xy graphene of 70% to 30%, whereas for H-xy graphene, these values are equal to 71%/29%, which is in complete agreement with the data derived from the NMR spectra. Thus, the proposed approach for the deconvolution of the C 1*s* spectra of graphene derivatives with the determined *BE*s of the C-O-C and C-OH peaks allowed one to assess the absolute and relative concentrations of the epoxide and hydroxyl groups.

Despite the difference in the basal-plane oxygen groups, the relative concentrations of the ketone and carboxyl groups are comparable. This assumes that the possible difference in the DOS of E-xy and H-xy graphenes arises from the effect of the basal-plane epoxide and hydroxyl groups. Furthermore, the almost-equal relative concentrations of ketones and carboxyls results in the effective smoothing of those VB features that arise from oxygen 2*p* orbitals. This is due to these features having sufficiently large half-widths and similar intensities, along with the fact that the features inherent to carbonyl edge groups are mainly located between the features inherent to carboxyl ones [34].

### 3.3. Probing of the E-xy and H-xy Graphene via X-ray Absorption Spectroscopy

To further figure out the fingerprints of the epoxide and hydroxyl groups in the XPS and XAS spectra, C *K*-edge XAS spectra of the materials under study were collected, being displayed in Figure 3c. All the spectra were characterized by the presence of the pronounced π* resonance centered at *hv* = 285.1–285.3 eV, which matured from the electron transitions in the C=C bonds at pristine *sp*^2^ domains [57]. In rGO, it was accompanied by a well-defined Frenkel σ*-exciton resonance centered at *hv* = 291.65 eV, which was excited sufficiently and reliably only if the size of the respective graphene domain exceeded ~five periods of its lattice and signified the complete recuperation of the conjugated graphene network [58,59]. The absence of this exciton in both the E-xy and H-xy graphene spectra indicated that the basal plane of these derivatives was sufficiently, densely, and uniformly covered with basal groups.

In turn, in the E-xy and H-xy graphene spectra, a set of resonances at *hv* = 286.2 and *hv* = 288.4 eV could be distinguished, which are commonly attributed to the C 1*s*-π* resonances of the electronic transitions in C-O bonds in the edge-located hydroxyls and C=O bonds in the ketone and carboxyl groups, respectively [52,57,60]. Besides these resonances present in both derivatives, the spectra of the E-xy and H-xy graphenes were manifested by their individual spectral features centered at *hv* = 287.3 eV and *hv* = 289.6 eV, respectively. The assignment of the peak at *hv* = 287.3 eV to the π*(C-O-C) resonances is generally accepted [51,60,61], although there are a number of data asserting that CK edge features with an energy of ~287.2–287.5 eV are to be assigned to the σ* resonance of C-H/C-H_2_ groups [62]. However, no signs of the presence of carbohydrate groups in the studied E-xy or H-xy graphenes were indicated in the FTIR spectra, although the corresponding bending and vibrational bands are known to clearly reveal themselves in the range of *v* = 2750–2900 cm^−1^ even at low concentrations of C-H/C-H_2_ [48,63]. Combined with the absence of a distinguishable resonance at *hv* = 287.3 eV in the H-xy spectrum, the discussed spectral feature was assumed to arise from C 1*s*-π* electronic transitions in the epoxide groups.

The nature of the π*-resonance-like maximum with an energy of *hv* = 289.6 eV exclusively presented in the pre-edge region of the H-xy spectrum was more unambiguous. Some authors assign it to the π* resonance corresponding to carboxyl groups [64], which is not consistent with a set of studies on the C *K*-Edge XAS spectra obtained for carboxylated graphenes [47,53,57]; others ascribe it to a band of Rydberg states of the C=O group [65]. However, this peak was absent in the E-xy graphene spectrum, which contained only slightly fewer ketone and carboxyl groups. Furthermore, the elimination of this peak upon the removal of basal-plane groups with a simultaneous rise in the number of carboxyls was indicated earlier [47]. Given these results, the peak at *hv* = 289.6 eV was assigned to correspond to the σ* resonance of the hydroxyl groups on the graphene basal plane, fingerprinting this oxygen group in the C *K*-Edge XAS spectra.

### 3.4. Examination of the Valence Band of E-xy and H-xy Graphene

By figuring out the chemistry of E-xy and H-xy graphene and verifying the predominance of either epoxide or hydroxyl groups in the synthesized derivatives, the examination of the VB of the acquired materials becomes feasible. Figure 4a exhibits the photoelectron spectra (PES) of the VB of E-xy and H-xy graphene as well as rGO collected using synchrotron radiation of *hv* = 130 eV and acquired at the normal angle of the photoelectron escape. All three spectra were quantitatively processed by the method of second derivatives, which allowed the significant refinement of the energy position of VB features that were only slightly visible in the original PES spectra of E-xy and H-xy graphene (Figure 4b). The resolved features were mainly designated by capital letters (*A*, *B*, *C*, …, etc.), the order of which corresponds to the direction of increasing binding energies. Only two peaks, with the lowest one, having a *BE* ~ 31 eV, visible in all three spectra and the highest one, centered at a *BE* of 3.2 eV, defined only in the VB spectrum of the rGO sample, were labeled in a distinct way. The former one, denoted as Na 2*p*, corresponded to trace amounts of sodium ions adsorbed on the studied graphene derivatives. In turn, the latter one was related to the highest critical point of the graphene band structure and is discussed below.

Furthermore, owing to the high degree of reduction, all the features in the rGO VB spectrum and its second derivative were related to the local maxima of the graphene DOS, which are located in certain high-symmetry points (*Г*, *M*, and *K*) of its hexagonal Brillouin zone and are commonly referred to as critical points of the graphene electron band structure [16]. The assignment of these spectrum features was based on both the experimental data published [17,22,23] and our previous results on the *VASP-DFT* calculations of the graphene DOS [34]. Thus, a comparative analysis of this spectrum with the ones of E-xy and H-xy graphenes would allow the categorization of spectral features related to the pristine graphene network and the introduced epoxide and hydroxyl groups.

Accordingly, the *X* and *F’* features of the rGO VB spectrum that were absent in the E-xy and H-xy spectra were related to the critical points of the graphene band structure, namely $M_{2u}^{-}$ and ${Г}_{1g}^{+}$ or the $Q_{2u}^{-}$(π) ones, respectively [23,66]. A set of other spectral features, namely *A* (*BE* ~ 5.0 eV), *B* (*BE* ~ 7.8 eV), *D* (*BE* ~ 13.5 eV), *E* (*BE* ~ 17.0 eV), and *F* (*BE* ~ 19.3 eV), can be resolved in all three spectra, suggesting their possible dual nature. Particularly, these spectral bands probably comprise both states of the π-conjugated carbon atoms forming domains of the pure graphene DOS maxima and the molecular-related states of the epoxide and hydroxyl groups, that is, the energy levels of molecular orbitals formed upon the covalent bonding of the carbon and oxygen atoms. Conversely, the features *C*, *G’*, and *G*, with *BEs* of 10.6 eV, 24.7 eV, and 26.9 eV, respectively, were absent in the VB spectrum of rGO and revealed themselves only in the spectra of the E-xy and H-xy graphenes. Hence, they can be unambiguously identified as oxygen-related ones.

The origin of the prominent and broad feature *G* is related to the O 2*s* shallow core level, participating in the formation of σ-type C-O orbitals of the presented oxygen functional groups. Owing to the difference in the composition and geometry of groups, the *BE* of this core level varied from one functionality to other, resulting in a broad peak summarizing the impact from the states of the C-O orbitals of all oxygen groups. This was reflected by the identification of its sub-component G’, revealing itself only in the second-derivative spectra, which corresponded to the C-O orbitals of carboxyl groups and was almost undetectable due to the low relative concentration of these oxygen moieties in the H-xy and E-xy graphene [34].

To further perform an accurate analysis of the *A*–*C* features, the rGO VB spectrum was normalized in such a way that the intensity of its high-energy tail (*BE* > 25 eV) became approximately equal to the intensities of the corresponding tails of spectra 1 and 2 (after the respective subtraction of an intensity coming from peak *G*). Thus, the acquired spectrum was overlaid on the VB spectra of E-xy and H-xy graphene (dotted line). Such an approach allowed the identification that the DOS in the region of *BE*s 5–11 eV was much higher in the VB spectra of E-xy and H-xy graphene compared to the rGO one. Since the probing depth and lateral dimensions of the X-ray beam were the same for all the samples, such a difference in the VB intensities points out that the *A-C* features were predominantly contributed to by electrons localized on the occupied oxygen 2*p* atomic orbitals (*AO*s), comprising *MO*s of the basal-plane hydroxyl and epoxide groups. This assertion is supported by the fact that the photon energy of 130 eV 2*p* subshell photoionization cross-section in the case of oxygen was about 7.5 times larger than one of carbon [67]. Thus, the intensity of the VB spectra regions related to the graphene network covered by oxygen groups was enhanced compared to the one of pristine graphene, being measured in otherwise identical conditions.

In the case of the *D*, *E*, and *F* features, a contrary situation was observed: the relative intensity of these peaks in the E-xy and H-xy spectra were either almost equal (feature *D*) or even noticeably lower (features *E* and *F*) than in the rGO VB spectrum. This evidenced that the main contribution to the intensity of these features was provided by photoelectrons emitted from the carbon atoms of the pristine graphene network. Thus, the features *D*, *E*, and *F* were assumed to be associated with electronic states of such critical points of the graphene band structure in this energy range as $K_{1}^{+}$(σ),$\;K_{3}^{+}$(σ), and $M_{1g}^{+}$(σ), respectively. Notably, this interpretation coincides with the published data on GO VB [17,18,66], although here it was proposed independently based on the examination of the acquired experimental data.

A further comparison of the E-xy and H-xy spectra, both in their original forms and in the form of second derivatives, also revealed differences in the DOS of these derivatives. Despite comprising the same sets of spectral features with almost identical *BE*s, the relative intensities and FWHM of the same features differed. Most obviously, this was illustrated by the *A* and *B* features.

### 3.5. Modeling Spatial and Energy Distribution of E-xy and H-xy Group-Related Electron Density

To assess the nature of the *A-G* features in the experimental spectra, DFT calculations of the graphene layer, functionalized by either a single epoxide or hydroxyl group, were further performed. Figure 5 and Figure 6 exhibit the total DOS (TDOS) calculated for the graphene layer modified by a single epoxide or hydroxyl group (curve 1) and the pristine graphene layer (curve 2). Apparently, the derived TDOS were practically the same due to the minor impact of a single oxygen group on the DOS of a graphene layer of 240 atoms. Thus, the energy distribution and spatial configuration of the electronic density (ED) surrounding the atoms of the modifying oxygen group were calculated in terms of the formalism of the projected density of electron states (pDOS). Within the frame of this approach, energy functions describing the energy spectra of the total DOS projected on the *AO*s of the oxygen and carbon atoms of either the epoxide or hydroxyl groups can be derived, and the energy distribution of the overall electron density localized at the examined atoms can be visualized. Given these results, one can identify whether the spectral feature in the examined range of *BE*s is related to a molecular-related state in the epoxide or hydroxyl group and, if yes, determine the type of the corresponding *MO*—nonbonding, which refers to lone pair electrons on the oxygen atom, or π or σ C-O bonding.

Thus, the derived pDOS spectra for the C, O, and H atoms along with their sum (total projected densities of states, TpDOS) are displayed in the bottom sections of Figure 5 and Figure 6. In turn, the upper sections present the visualizations of the ED localized on the *AO*s of oxygen, carbon, and hydrogen atoms in the vicinity of the epoxide or hydroxyl. To accurately derive the corresponding EDs, the whole energy range was divided into nine regions denoted as (B0, B1, …, B8). Regions B0 and B7 were not related to the experimentally indicated spectral features, but were discerned since they were presented by a particular ED configuration. At the same time, the other regions corresponded to a particular experimental spectral feature (*A*, *B*, C, … etc.), being established in such a way that the corresponding spectral feature (*A* for B1, *B* for B2, … *G* for B8) laid approximately in the center of the corresponding region, and the regions had close widths, while the overlap of the regions, corresponding to different ED configurations, would be absent.

As seen, the B0 region did not contain any pronounced maximum except intensive onset, which completely came from the OpDOS component. According to the respective images of the ED distributed within the B0 region in both the E-xy and H-xy graphenes, the corresponding electronic states apparently arose from *MOs* occupied by oxygen 2*p* nonbonding lone pair electrons n’O2*p* (HOMO). Moving to higher *BE*s, the dominant maximum in the OpDOS component was indicated for both the E-xy and H-xy graphenes. Based on the analysis of the ED spatial distribution, the states in this energy range were attributed to 2*p*-type oxygen *AO*, oriented parallel to the basal plane of the graphene layer.

Despite the similarity in the band structure of E-xy and H-xy graphene in the B0 and B1 regions, the following B2-B4 regions for these graphene derivatives had a distinct structure. The nature of the maximum in the B3 region was asserted to be related to a system of covalent σ bonds between atoms of the graphene layer and oxygen, constituting the C-O-C heterocycle of the epoxide group. This was seen from the respective ED pattern, demonstrating a high electron density surrounding the O and C atoms and filling the space between them. This was furthermore supported by the fact that corresponding electron density was not concentrated on one certain energy level, as in the case of molecular-like bonding, but distributed over a sufficiently wide band of energy levels of a similar nature as in the case of solid-state-like bonding. Particularly, the electron density, which participates in the formation of σ bonding between all three atoms of the E-xy group, continued to retain its spatial configuration at least in two neighboring regions to B3, that is, in the B2 and B4 ones.

Unlike the case of the E-xy graphene, the TpDOS of the H-xy graphene exhibited three rather pronounced maxima located in each of the B2-B4 regions. The comparative analysis of the displayed pDOS showed that the maxima in the TpDOS of the B2 and B4 regions coincided with the maxima in the HpDOS, while in the case of the B3 region, no contribution came from HpDOS. Accordingly, the states localized in the B2 and B4 regions were considered to be related to the *MO*s of the σ(O-H) bonding in the hydroxyl groups, whereas the ED in the B3 region mainly stemmed from the σ(C-O) bonding between the group and the graphene layer. This assertion is well supported by the respective ED spatial configurations, for which a high electron density around the hydrogen atom was indicated in the case of the B2 and B4 regions, with its absence in the case of the B3 energy range.

Moving to the B5-B7 regions, the apparent diminishment of the OpDOS was indicated, with a minor impact of the ED localized on the O atoms, verifying that the DOS in this energy region was mainly constituted by the states of the graphene network. It is worth noting that the ED corresponding to the graphene domains was distributed uniformly in a layer, covering all the carbon atoms that comprised the supercell. As a result, it was extremely difficult to visualize its density maxima coinciding with the critical points of the graphene band structure. On contrary, the maxima of the ED lying outside this layer, i.e., the ED corresponding to the oxygen and carbon *AO*s, namely C2*p_z_ AO* for the latter ones, were visualized quite easily. For this reason, the relatively weak maxima located for the TpDOS of both groups in the B5 and B6 regions practically did not manifest themselves visually. Moreover, the ED images corresponding to these regions showed a significant weakening of the covalent bonds between all atoms of the group, simply due to a noticeable decrease in the amount of the ED participating in the formation of these bonds.

Finally, a drastic and narrow rise in the TpDOS, driven mainly by the OpDOS, was observed in the B8 region, which laid outside the range of the graphene VB electron states. Owing to this fact and based on both the comparison of the pDOS and the respective ED visualization, the corresponding electronic states were assigned to the shallow core levels corresponding to the σ(C-O) bond formed mostly by the overlapping of the O 2*s* and C 2*s AO*s.

Thus, the performed theoretical calculations fully verified the relevance of the interpretation of the VB spectra modification and the appearance of the *A*, *B*, *C*, and *G*(*G*’) upon functionalization by epoxides and hydroxyls in terms of the introduction of the molecular states related to the introduced functional groups. Particularly, the *A* spectral feature was related to the states of the 2*p*-type oxygen *AO* of the modifying groups, whereas the B and C spectral features contributed to the states assigned to the C-O σ bonding, both in the E-xy and H-xy graphene, and additionally to the O-H σ bonding in the latter case. Furthermore, the fact that the G spectral feature is related to the O 2*s* shallow core level, which participates in the C-O σ-bonding, has been additionally verified by the performed calculations. The absence of any signs of the molecular-related states of the 2*p* nonbonding lone pair electrons in the range of 0–1 eV in the experimental VB spectra was attributed to the adsorption of the water molecules. Owing to the lone pair electrons of the hydroxyls and epoxides participating in the hydrogen bonding, the corresponding states diminished, as was demonstrated earlier [53,68]. Table 2 summarizes the provided assignment for the distinguished VB features with a comparison to the literature data.

## 4. Conclusions

To summarize, advanced features of examining graphene functionalized by epoxide and hydroxyl groups via the core-level methods were signified. The fingerprints of these oxygen groups in the C *K*-edge XAS spectra were refined, whereas the deconvolution of the graphene derivative C 1*s* spectra with the discerned spectral components related to the epoxide and hydroxyl groups was proposed and justified by comparing to the FTIR and NMR spectroscopic data. Given this approach, the relative and absolute concentrations of exclusively an epoxide or hydroxyl group, not just their sum, can be derived from the acquired C 1*s* spectra. This is crucial for a set of strategies for the synthesis of graphene-based bio- and nanocomposites, in which either an epoxide ring or hydroxyl group is required for the reaction to proceed. Furthermore, such processing of the C 1*s* spectra of the GO and rGO layer grants an accurate estimation of the functionalization degree independently by epoxides or hydroxyls, which is of high interest for both fundamental studies and the practical application of these materials in the field of electrochemistry.

Besides chemistry analysis, the peculiarities of the graphene VB alteration upon introduction of the epoxide and hydroxyl groups were specified. The appearance of localized states related to the *MO*s of the modifying functionalities was shown and identified by applying DFT calculations by means of using a projection of the total density of states. The applied approach allowed the decomposition of the impact of the localized states corresponding to non-bonding lone pair electrons of n_O2*p*_, σ, and π bonds. The presented results highlight graphene functionalization by an epoxide or hydroxyl group to be an advanced tool for engineering its deep electronic states, complementing the common studies on the basal-plane functionalization for the alteration of its bandgap. Moreover, the performed examination of the VB spectra can be regarded as a powerful tool to additionally elucidate the chemistry of graphene materials.

Taken together, these findings guide and extend the frames of studying graphene derivatives and engineering their band structure for practical applications.

## Figures and Tables

**Figure 1 nanomaterials-13-00023-f001:**
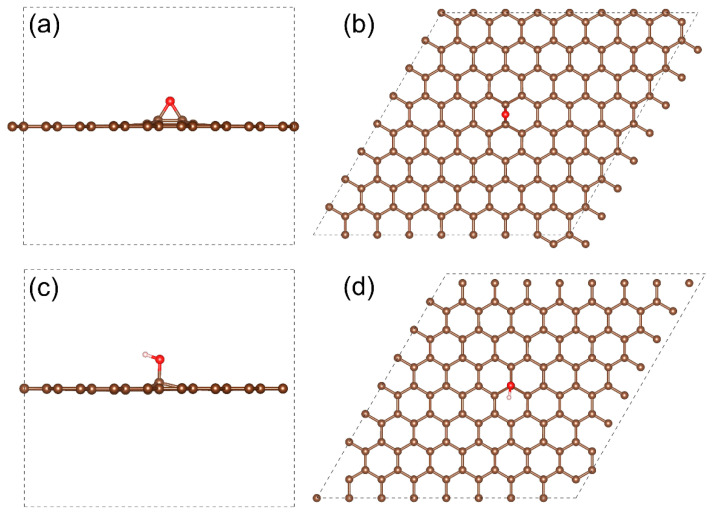
The supercells of the (**a**,**b**) E-xy graphene and (**c**,**d**) H-xy graphene taken for the DFT studies, top and side views.

**Figure 2 nanomaterials-13-00023-f002:**
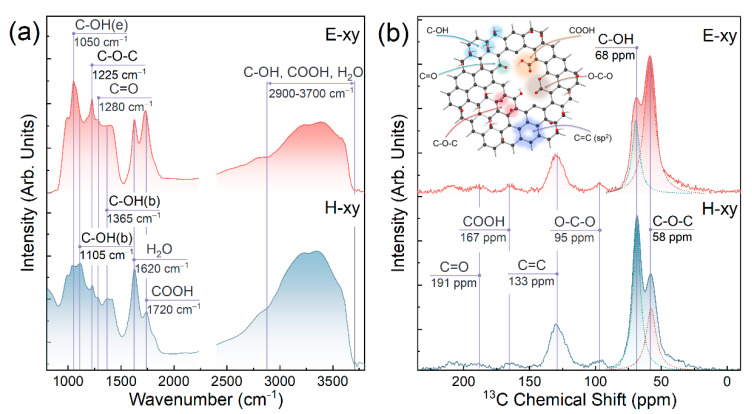
(**a**) FTIR and (**b**) NMR characterization of E-xy and H-xy graphenes. Inset—schematic representation of the GO chemistry.

**Figure 3 nanomaterials-13-00023-f003:**
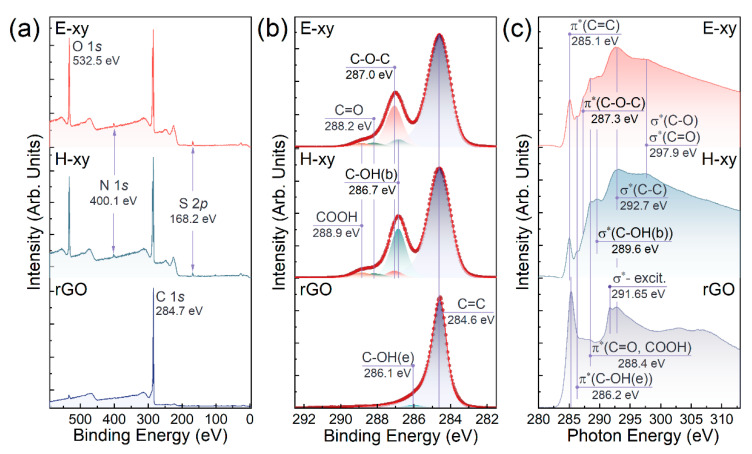
X-ray photoelectron examination of the CMG chemistry. (**a**) Survey and (**b**) core-level C 1*s* X-ray photoelectron spectra of the E-xy graphene, H-xy graphene, and rGO. (**c**) C *K*-edge X-ray absorption spectra of the investigated materials.

**Figure 4 nanomaterials-13-00023-f004:**
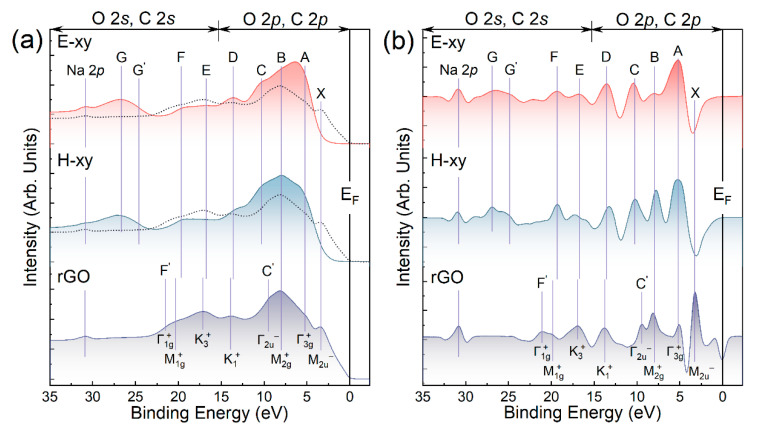
The original (**a**) and second-derivative (**b**) VB photoemission spectra acquired for E-xy and H-xy graphene along with rGO film.

**Figure 5 nanomaterials-13-00023-f005:**
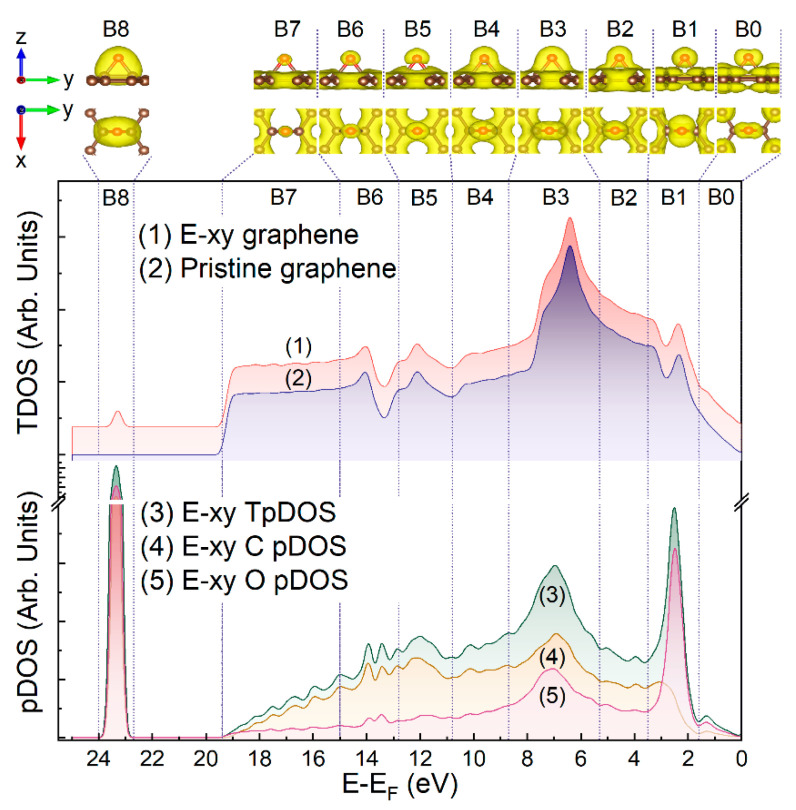
The calculated DOS of the E-xy graphene, pristine graphene layer, C pDOS, O pDOS, and TpDOS, and images of the electron distribution for the epoxide group on the graphene layer.

**Figure 6 nanomaterials-13-00023-f006:**
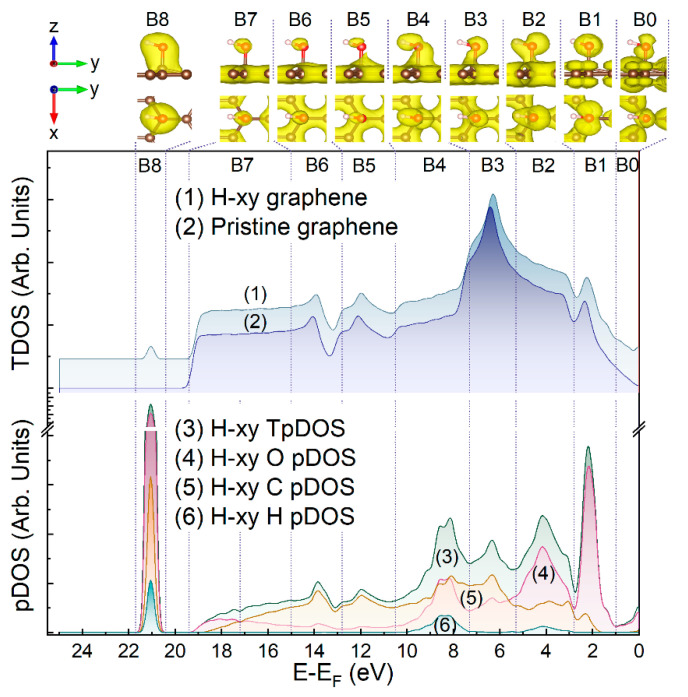
The calculated DOS of the H-xy graphene, pristine graphene layer, C pDOS, O pDOS, H pDOS, and TpDOS, and images of the electron distribution for the hydroxyl group on the graphene layer.

**Table 1 nanomaterials-13-00023-t001:** Relative concentrations of the main four functional groups determined for E-xy and H-xy graphene samples.

Component	C=C	C-OH	C-O-C	C=O	COOH	C-O-C/C-OH Ratio
Binding Energy (eV)	284.6	286.8	287.0	288.2	288.9	
E-xy	72.4	3.7	19.5	2.4	2.0	70%/30%
H-xy	69.4	21.7	3.2	2.8	2.5	29%/71%

**Table 2 nanomaterials-13-00023-t002:** Comparison of the VB feature assignments for the studied graphene derivatives.

VB Feature	E−E_F_ (eV)	Assignment					
		This Work	[16]	[17]	[18]	[23]	[66]
		E-xy and H-xy Graphene	GO/Graphite	GO/rGO	GO/rGO	HOPG	Graphene
X	3.2	${Г}_{1g}^{+}$$(\pi )/Q_{2u}^{-}$(π) (C=C)	$Q_{2u}^{-}$(π) (C=C)	${Г}_{1g}^{+}$(π) (C=C)	-	$Q_{2u}^{-}$(π) (C=C)	${Г}_{1g}^{+}$(π) (C=C)
A	5.0	n’(O2p)	${Г}_{3g}^{-}$(σ) (C=C)	-	-	-	-
B	7.8	σ(C-O)	$(C=C)\;{Г}_{2u}^{-}$ & O2*p*-states	-	-	${Г}_{2u}^{-}$ (C=C)	-
C’	9.5	${Г}_{2u}^{-}$(σ) (C=C)	$Q_{2g}^{+}$(σ) (C=C)	-	-		-
C	10.6	σ(C-O)/σ(O-H)		-	-		-
D	13.5	$K_{1}^{+}$(σ) (C=C)	$Q_{1u}^{+}$(σ) (C=C)	$K_{1}^{+}$(σ) (C=C)	$K_{1}^{+}$(σ) (C=C)	$K_{3}^{+}$ (C=C)	$K_{1}^{+}$(σ) (C=C)
E	17.0	$K_{3}^{+}$(σ) (C=C)	$Q_{1g}^{+}$(σ) (C=C)	$K_{3}^{+}$(σ) (C=C)	$K_{3}^{+}$(σ) (C=C)	-	$K_{3}^{+}$(σ) (C=C)
F	19.3	$M_{1g}^{+}$(σ) (C=C)	${Г}_{1g}^{+}$(σ) (C=C)	$M_{1g}^{+}$(σ) (C=C)	$M_{1g}^{+}$(σ) (C=C)	-	$M_{1g}^{+}$(σ)(C=C)
F’	21.5	${Г}_{1g}^{+}$(σ) (C=C)		$K_{1}^{+}$(σ) (C=C)	$K_{1}^{+}$(σ) (C=C)	-	$K_{1}^{+}$(σ) (C=C)
G	24.7	O2*s AO*s of σ(C-O)	O2s states	-	-	-	-
G’	26.9	O2*s AO*s of σ(COOH)		-	-	-	-

## Data Availability

The data presented in this study are available from the first author upon request.

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
