# Peer review of "Manifesting Epoxide and Hydroxyl Groups in XPS Spectra and Valence Band of Graphene Derivatives"

_nanomaterials, 2022, doi:10.3390/nano13010023_

Round 1
Reviewer 1 Report
The manuscript describes a combined experimental and computational study of the changes induced by O functionalization in the core-level spectroscopic properties of graphene. The authors use a combination of spectroscopic techniques (XPS, XAS, FTIR and NMR) along with density-functional theory (DFT) calculations to identify the fingerprints of epoxidized and hydroxylated graphene and to interpret their peak structure in terms of molecular orbitals of the O groups. They show that the relative and absolute concentration of hydroxyl and epoxide groups can be determined from the measured C 1s spectra. They also study in detail the effects of functionalization on the distribution of electronic states within the valence band of graphene, and they link specific spectral features to molecular orbitals localized on the OH and COC groups and to the nature of the bonds established between the O groups and the graphene carbon atoms.
The work is well carried out and the results are well supported by the experimental and computational data presented. The use of DFT calculations as an interpretative tool for experimental data is shown to provide unique insight into the nature of the interactions between hydroxyl/epoxide groups and the graphene framework. This study is important for future work aimed at engineering the electronic properties of graphene using chemical modification. I recommend publication, but I would like the authors to also address the following minor points.
1) In figure 4 and 5, what is the meaning of "Binding Energy"? How were these biding energies computed from the DFT calculations? What is the zero of binding energy in these plots? This should be explained in the paper.
2) In the caption of Figure 5 "H-xy graphene" should replace "E-xy graphene".
3) The interpretation of DOS features in the B0-B7 region may be complicated by the fact the DFT, particularly in its local density approximation (LDA), tends to overdelocalize electrons. Therefore, the actual relative energies of orbitals containing a localized component may not be accurate. Is there a specific reason why the authors decided to use a relatively poor approximation for the exchange-correlation functional of DFT insteadcof approximations that may be able to provide more accurate energies of localized states relative to the delocalized pi system of graphene?
Reviewer 2 Report
In this paper, the authors proposed advanced fingerprints of epoxides and hydroxyl groups on graphene layer by core layer method, and revealed the modification of their valence bands (VB) after the introduction of these oxygen functional groups. The distinctive contribution of epoxide and hydroxyl groups to the C 1s X-ray photoelectron spectra is indicated experimentally, allowing to quantitatively characterize each group, not just their sum. This work reveals the special contribution of epoxides and hydroxyl groups to the core energy level spectrum and energy band structure of graphene derivatives, promotes the functionalization of graphene, and uses it as a tool for engineering physical properties. I believe that publication of the manuscript may be considered only after the following issues have been resolved.
1. In order to better highlight the advantages of this work, the author needs to provide a table to compare related work.
2. Part 2.3 of the article, as it is a routine test, does not need to be too detailed. The author needs to simplify this part.
3. In the DFT Modeling part, the author needs to provide a specific schematic diagram of the model.
4. For references 63 and 64, it is recommended that the author place them in the appropriate part of the text.
5. The introduction can be improved. The articles related to some applications of graphene materials should be added such as Sensors 2022, 22, 6483; ACS Sustain. Chem. Eng. 2015, 3, 1677–1685; RSC Adv. 2022, 12, 7821–7829; Talanta 2015, 134, 435–442.
6. Please check the grammar and spelling mistakes of the whole manuscript.
Round 2
Reviewer 2 Report
Accept in present form.